# CC2D1A causes ciliopathy, intellectual disability, heterotaxy, renal dysplasia, and abnormal CSF flow

Angelina Haesoo Kim[1,]*, Irmak Sakin[2,3,]*, Stephen Viviano[1], Gulten Tuncel[4], Stephanie Marie Aguilera[5], Gizem Goles[5], Lauren Jeffries[6], Weizhen Ji[6], Saquib A Lakhani[1,6], Canan Ceylan Kose[7], Fatma Silan[7], Sukru Sadik Oner[8,9], Oktay I Kaplan[10], MarmaRare Group[11,]†, Mahmut Cerkez Ergoren[12], Ketu Mishra-Gorur[13], Murat Gunel[13,14,15,16], Sebnem Ozemri Sag[17], Sehime G Temel[17,18,]*, Engin Deniz[1,6,]*

**Intellectual and developmental disabilities result from abnormal nervous system development. Over a 1,000 genes have been associated with intellectual and developmental disabilities, driving continued efforts toward dissecting variant functionality to enhance our understanding of the disease mechanism. This report identified two novel variants in *CC2D1A* in a cohort of four patients from two unrelated families. We used multiple model systems for functional analysis, including *Xenopus*, *Drosophila*, and patient-derived fibroblasts. Our experiments revealed that *cc2d1a* is expressed explicitly in a spectrum of ciliated tissues, including the left–right organizer, epidermis, pronephric duct, nephrostomes, and ventricular zone of the brain. In line with this expression pattern, loss of *cc2d1a* led to cardiac heterotaxy, cystic kidneys, and abnormal CSF circulation via defective ciliogenesis. Interestingly, when we analyzed brain development, mutant tadpoles showed abnormal CSF circulation only in the midbrain region, suggesting abnormal *local* CSF flow. Furthermore, our analysis of the patient-derived fibroblasts confirmed defective ciliogenesis, further supporting our observations. In summary, we revealed novel insight into the role of *CC2D1A* by establishing its new critical role in ciliogenesis and CSF circulation.**

## Introduction

Intellectual and developmental disorder (IDD) is the result of abnormal early development of the central nervous system that clinically presents with intellectual and adaptive functioning deficits in conceptual, social, and practical domains [1]. Approximately 2% of the world's population is affected by IDDs, which remains the most common reason for referral to genetic analysis [2, 3, 4]. Pathogenesis of the disease is complex; over a 1,000 genes have been linked to non-syndromic IDD, suggesting a broad spectrum of molecular defects presenting with intellectual disorders [3, 5]. To close this gap in our knowledge, ongoing work is focused on the functional analysis of the variants and a better understanding of the disease mechanism.

The coiled-coil and C2 domain–containing protein 1A (*CC2D1A*) (OMIM#610055) has been implicated in a wide range of pathways, including NF-κB enhancer binding protein signaling [6, 7], phosphodiesterase activity [8], and regulation of serotonin 1A, dopamine D2 receptor [8, 9, 10, 11], bone morphogenetic protein [12], and Notch signaling [13]. In the cytoplasm, it acts as a scaffold protein in the PI3K/PDK1/AKT pathway and centrosome cleavage, mediating centriole cohesion during mitosis [14]. In vivo studies in mice showed that *CC2D1A* is widely expressed in the brain [15], and the *cc2d1a*-deficient mice died immediately after birth, indicating an essential role during early development [16]. Although the conditional brain-specific knockout mouse models showed cognitive deficits [17] and autistic-like phenotypes when *CC2D1A* was lost in

[1]Department of Pediatrics, Yale School of Medicine, New Haven, CT, USA   [2]Department of ENT, Cambridge University Hospitals NHS Foundation Trust, Cambridge, UK   [3]Acibadem University School of Medicine, Istanbul, Turkey   [4]DESAM Research Institute, Near East University, Nicosia, Cyprus   [5]Department of Neurosurgery, Yale University School of Medicine, New Haven, CT, USA   [6]Pediatric Genomics Discovery Program, Department of Pediatrics, Yale University School of Medicine, New Haven, CT, USA   [7]Canakkale 18 March University, Faculty of Medicine, Department of Medical Genetics, Canakkale, Turkey   [8]Department of Pharmacology, Goztepe Prof. Dr. Suleyman Yalcin City Hospital, Istanbul, Turkey   [9]Istanbul Medeniyet University, Science and Advanced Technologies Research Center (BILTAM), Istanbul, Turkey   [10]Rare Disease Laboratory, School of Life and Natural Sciences, Abdullah Gul University, Kayseri, Turkey   [11]MarmaraRare Group, Istanbul, Turkey   [12]Department of Medical Genetics, Faculty of Medicine, Near East University, Nicosia, Cyprus   [13]Department of Neurosurgery, Yale School of Medicine, New Haven, CT, USA   [14]Department of Genetics, Yale University School of Medicine, New Haven, CT, USA   [15]Yale Program in Brain Tumor Research, Yale University School of Medicine, New Haven, CT, USA   [16]Department of Neuroscience, Yale University School of Medicine, New Haven, CT, USA   [17]Department of Medical Genetics, Faculty of Medicine, Uludag University, Bursa, Turkey   [18]Department of Histology and Embryology and Health Sciences Institute, Department of Translational Medicine, Faculty of Medicine, Bursa Uludag University, Bursa, Turkey

Correspondence: engin.deniz@yale.edu; sehime@uludag.edu.tr
*Angelina Haesoo Kim, Irmak Sakin, Sehime G Temel, and Engin Deniz contributed equally to this work
†MarmaRare Group members are listed in the below Appendix.

 

**Table 1. Summary of *CC2D1A* variants and clinical features of the affected individuals.**

| Nucleotide | Protein | Reported phenotype | Reference |
|---|---|---|---|
| c.347delA | p.(Lys116Argfs*82) | Autism spectrum disorder, intellectual disability, and seizures | Manzini et al (23) |
| c.380C>T | p.(Pro27Leu) | Mental retardation, autosomal recessive | Xiong et al (30) |
| c.575C>T | p.(Pro192Leu) | Heterotaxy | Ma et al (31) |
| c.748 + 1G>T | (splice impacted) | Autism spectrum disorder, intellectual disability, and seizures | Manzini et al (23) |
| c.811delG | p.(Ala271Profs*30) | Intellectual disability, non-syndromic | McSherry et al (32) |
| c.1016delC | p.(Thr339Lysfs*95) | Developmental disorder | Abolhassani et al (33) |
| c.1345G>A | p.(Val449Met) | Neurodevelopmental delay | Jauss et al (34) |
| c.1517A>G | p.(Gln506Arg) | Heterotaxy | Ma et al (31) |
| c.1552G>T | p.(Glu518*) | Autism spectrum disorder | Iossifov et al (24) |
| | | | Lim et al (25) |
| | | | Turner et al (26) |
| | | | Fu et al (27) |
| | | | Zhou et al (28) |
| c.1591G>A | p.(Glu531Lys) | Congenital heart defects | Meerschaut et al (35) |
| c.1595C>T | p.(Pro532Leu) | Heterotaxy | Ma et al (31) |
| c.1610C>T | p.(Ser537Leu) | Mental retardation, autosomal recessive | Kamil et al (36) |
| c.1641 + 1G>A | (splice impacted) | Intellectual disability | Rashvand et al (37) |
| c.1647G>T | p.(Pro549=) | Autism | Zhou et al (28) |
| c.1739C>T | p.(Thr580Ile) | Smith–Magenis syndrome–like disorder, Joubert-like disorder | Loviglio et al (38) |
| | | | Tuncel et al (22) |
| c.2342G>A | p.(Gly781Glu) | Heterotaxy | Ma et al (31) |
| c.2342G>T | p.(Gly781Val) | Intellectual disability, heterotaxy | Wang et al (39) |
| | | | Ma et al (31) |
| c.2520-1G>T | (splice impacted) | Autism spectrum disorder | Sener et al (29) |
| c.2657G>A | p.(Arg886His) | Smith–Magenis syndrome–like disorder | Loviglio et al (38) |
| c.2693delG | p.(Gly898Valfs*45) | Intellectual disability, non-syndromic | Reuter et al (40) |
| c.2728G>A | p.(Glu910Lys) | Mental retardation, autosomal recessive | Xiong et al (30) |

glutamatergic neurons (18), similar to homozygous mice, they also died after birth, indicating potential additional roles of *CC2D1A* during embryonic development. An additional mouse study involving conditional postnatal removal of *CC2D1A* specifically in the forebrain revealed morphologically abnormal cortical dendrite organization and reduced density of dendritic spines, resulting in mice with deficient neuronal plasticity, spatial learning, and memory alongside features of reduced sociability, hyperactivity, anxiety, and excessive grooming (19).

Various phenotypes have been associated with *CC2D1A*, also known as *Freud-1*, *Lgd2*, and *Aki-1*, in the medical literature, but not all studies include detailed phenotyping. Consistent with data from murine modeling, the most well-characterized human patients have neurological phenotypes. Homozygous pathogenic variants, largely putative null alleles, have been associated with autosomal recessive non-syndromic intellectual disability (OMIM# 608443) (15,

20, 21, 22) (Table 1). Several groups also have identified recessive and dominant *CC2D1A* variants with various predicted molecular consequences in different cohorts of patients with autism spectrum disorder (ASD), with or without intellectual disability or seizures (21, 23, 24, 25, 26, 27, 28, 29).

Interestingly, Ma et al recently reported patients with congenital heart disease presenting with damaging mutations in the *CC2D1A* gene (31). The group identified seven damaging exonic missense variants of *CC2D1A* in six patients with congenital heart disease consistent with heterotaxy using whole-exome sequencing and recapitulated heterotaxy in the zebrafish model system (31). Heterotaxy is caused by defective left–right (LR) patterning, where one or more organs are misplaced along the left–right body axis where cilia play a pivotal role (41, 42).

Cilia are membrane-bounded projections from the cell surface and can be either motile, beating to generate extracellular fluid

flow (43), or immotile, acting as signaling centers (44, 45). Both types of cilia are upstream of diverse biological processes in human physiology, cell signaling, and embryonic development. Thus, the diseases featuring heterotaxy are often associated with defective ciliogenesis and are therefore commonly termed ciliopathies. The role of *CC2D1A* in ciliary biology and how that relates to intellectual disability remain to be determined.

This report shows that *CC2D1A* is essential for ciliogenesis and plays a pivotal role in multiple developmental processes regulated by cilia. We broaden the clinical spectrum linked to the *CC2D1A* gene by presenting a new clinical presentation, cystic renal disease, and show that *CC2D1A* is essential for nephrogenesis. To date, only six variants in *CC2D1A* have been reported in ClinVar to be pathogenic or likely pathogenic in patients presenting with intellectual disability (ClinVar), and only around two dozen variants have been clinically reported in the Human Gene Mutation Database.

Here, we report two additional nonsense novel variants. A novel homozygous (c.1186C>T [p.Arg396*]) variant in the *CC2D1A* gene was identified in two siblings with intellectual disability, autistic features, renal cyst, and obesity. Such a constellation of comorbidities has been previously included in ciliopathies (46). A second family with another novel homozygous (c.1264C>T [p.Glu422*]) variant was identified in two siblings presenting with intellectual disability, seizure disorder, and multiple renal cysts.

We used three distinct strategies to understand the role of *CC2D1A* in development and disease. (1) Using the frog *Xenopus tropicalis* model system, we revealed that *cc2d1a* is expressed explicitly in a spectrum of ciliated tissues, including the gastrocoel roof plate (GRP, left-right organizer), epidermis, pronephric duct, nephrostomes, otic vesicle, and ventricular zone of the brain. We showed that *CC2D1A* is localized to the base of the cilia in the GRP monociliated cells and the epidermal multiciliated cells (MCCs). Tadpoles developed abnormal left–right patterning and cystic kidneys when we depleted *cc2d1a* using the CRISPR/Cas9 system. Interestingly, when we analyzed brain development, mutant tadpoles showed abnormal cilium-driven CSF circulation, specifically in the midbrain region. Using immunohistochemistry (IHC) and scanning electron microscopy imaging, we showed that the midbrain ependymal motile cilia decorating the brain ventricle were locally disrupted. We also generated fibroblasts from our patients and demonstrated defective ciliogenesis. (2) We tested the specific patient variant by rescue experiments where WT human *CC2D1A* RNA injection rescued the left–patterning defect. In contrast, mutant RNA with the patient variant failed, indicating that the patient variant is indeed disease-causing. (3) Finally, to determine how *CC2D1A* affects social behavior because the null and brain-specific knockout mice died shortly after birth, we used the *Drosophila* model and well-established assays to study social behavior (47). Indeed, *CC2D1A* mutant fruit flies recapitulated the antisocial behavior our patients presented with.

In summary, our report expands on the role of *CC2D1A* in normal development and its associated disease, which includes intellectual and developmental disability, congenital heart disease, and cystic kidney disease, establishing its essential role in ciliogenesis.

# Results

## Clinical summaries and genetic analysis

We identified four patients from two unrelated families in different parts of Turkey presenting with different homozygous variants in the *CC2D1A* gene. Detailed clinical descriptions are presented in Table 1. *CC2D1A* is highly conserved across the species; both mutations were in proximity (Fig 1). To summarize, in Family 1, a 16-yr-old female patient presented with symptoms of intellectual disability (ID), obesity, and ASD. Similarly, her 12-yr-old male sibling presented with ASD and intellectual disability but in addition had a dysplastic and dysfunctional left kidney harboring multiple renal cysts. The siblings were born to healthy, consanguineous parents of Turkish heritage by normal vaginal delivery with unremarkable birth history. Both siblings carried a novel homozygous c.1186C>T (p.Arg396*) nonsense variant inherited from each of the heterozygous parents (Fig 1A and C). In Family 2, eight children were born to healthy, consanguineous parents of Turkish heritage. Both parents had heterozygous *CC2D1A* variants confirmed with Sanger sequencing. Of the eight children, two siblings, with no significant prenatal history, were referred to our clinic because of severe intellectual disability and seizure disorder. Both siblings carried a novel homozygous c.1264C>T (p.Glu422*) nonsense variant inherited from each of the heterozygous parents (Fig 1B and C). One sibling, a 17-yr-old male, also demonstrated dysmorphic facies and was diagnosed with ASD. The other sibling, a 25-yr-old female, in addition to severe intellectual disability and seizure disorder, presented with renal cysts, multiple small cysts on the right kidney, and a 15-mm parapelvic–cortical left renal cyst. She was also identified to have a de novo missense variant c.11257C>T (p.Arg3753Trp) (NM_001009944) in *PKD1* (polycystin 1), a gene known to play a role in kidney development and lead to autosomal dominant polycystic kidney disease (48). This de novo mutation showed 0% frequency in gnomAD and extremely low frequency in other population databases. Segregation analysis was performed for Family 2 (parents and male sibling), which confirmed the *PKD1* variant to be de novo and carried only by the affected female sibling. The results were confirmed with tissue and blood Sanger sequencing.

## *Cc2d1a* knockdown leads to dysplastic kidneys

Two of our patients from two different families presented with renal cysts. The renal USG of the male patient (Family 1—12 yo) at 8 mo of age demonstrated the cortical renal cyst (four cystic lesions with dimensions of 18 × 12.2 mm, 14.9 × 14.8 mm, 23.5 × 14.6 mm, and 9 × 11 adjacent to the upper pole of the left kidney were reported—Table 2), and the follow-up DMSA study at 2 yr of age demonstrated dysplastic, dysfunctional left kidney (Fig 1D). The renal USG of the female patient (Family 2—25 yo) showed multiple bilateral cystic lesions (Fig 1E), significantly more aggressive than our male patient's phenotype.

We investigated the impact of *cc2d1a* depletion on renal development using the frog *Xenopus* model system. To begin our studies, we used the whole-mount in situ hybridization to understand the expression pattern of the *cc2d1a* during renal

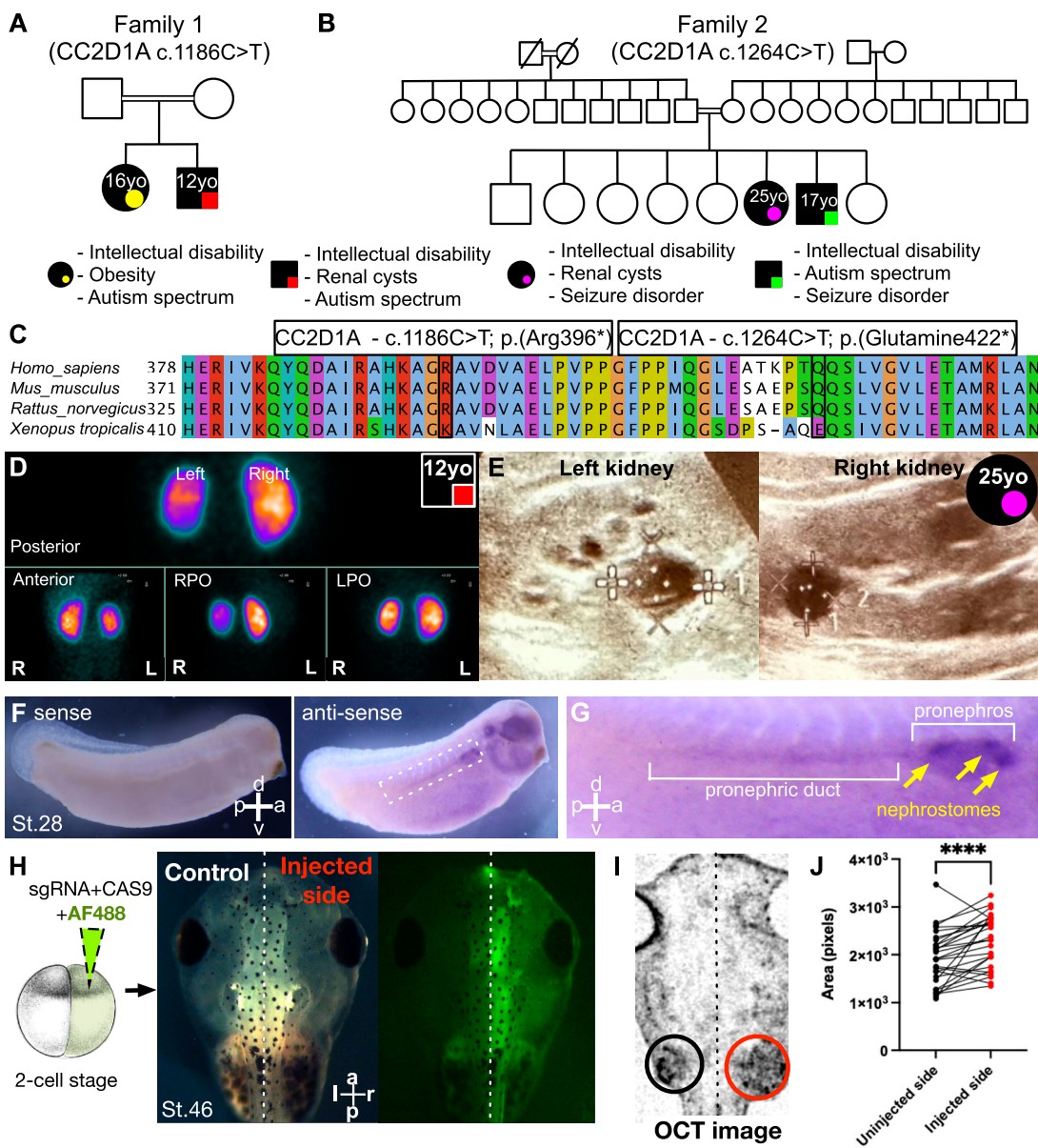

**Figure 1. cc2d1a knockdown leads to dysplastic kidneys.**
**(A, B)** Pedigrees and clinical presentation of four individuals from two unrelated families with homozygous mutations in CC2D1A, presenting with intellectual disability, autism, obesity, renal cysts, and/or seizures. Symbols are as follows: filled, affected; empty, unaffected; circle, female; square, male; yellow circle, 16-yr-old female from Family#1; red square, 12-yr-old male from Family#1; purple circle, 15-yr-old female from Family#2; green square, 17-yr-old male from Family#2; double bars, consanguinity in the family; hash, deceased. Abbreviations: yo, year old. **(C)** Alignment of the CC2D1A amino acid sequence from the human, mouse, rat, and frog shows high conservation across all species tested. The locations of the identified mutations are shown. **(D)** Renal ultrasonography of the 12-yr-old male patient from Family#1 at 8 mo of age shows cortical renal cysts (four cystic lesions). l, left; r, right; RPO, right posterior oblique; LPO, left posterior oblique. **(E)** Renal ultrasonography of the 25-yr-old female patient from Family#2 shows bilateral cystic lesions. **(F)** Whole-mount in situ hybridization on WT *X. tropicalis* embryos at st.28 with antisense (left image) and sense (right image) probes against *cc2d1a* mRNA. a, anterior; p, posterior; d, dorsal; v, ventral; st., stage. **(G)** Zoomed-in image shows strong *cc2d1a* expression in the pronephric duct and pronephros. Yellow arrows point to the nephrostomes, which are precursors to the kidneys. **(H)** Schematic of a one-of-two cell injection with *cc2d1a* sgRNA, Cas9, and Dextran, Alexa Fluor 488 dye. The embryos were raised to st.46, and with fluorescence microscopy, we verified that only one side was injected (shown in green). The non-fluorescent side serves as the internal control. r, right; l, left; a, anterior; p, posterior. **(I)** Representative 3D image of a st.46 *Xenopus* embryo, ventrally imaged using OCT. The right side, depleted of *cc2d1a*, exhibited larger kidneys (as shown in the red circle). OCT, optical coherence tomography. **(J)** Quantification of kidney cross-sectional area in the uninjected versus injected side in n = 28 embryos. Red and black dots indicate injected and uninjected side embryos, respectively. Paired *t* test; **** indicates *P* < 0.0001.

development. Indeed, at stage 28 (post-fertilization day 3 at 26°C), *cc2d1a* expression was strong in the pronephros and pronephric duct (Fig 1F). Specifically, the three punctuate patterns of *cc2d1a*

expression correspond to the nephrostomes, which are ciliated peritoneal funnels that connect the celomic cavity to the nephron (49, 50), and the precursors to the kidneys before organogenesis

**Table 2.** Summary of *CC2D1A* high-confidence variants.

| Pedigree | Family 1 | | Family 2 | |
|---|---|---|---|---|
| Age (yr) | 16 | 12 | 25 | 17 |
| Gender | Female | Male | Female | Male |
| Birth | 39 wk, vaginal delivery | 39 wk, vaginal delivery | | |
| Geographic origin | Turkish | Turkish | Turkish | Turkish |
| Parental consanguinity | + | + | + | + |
| CC2D1A variant | c. 1186C > T | c. 1186C > T | c. 1264C > T | c. 1264C > T |
| | p. (Arg396*) | p. (Arg396*) | p. (Glutamine422*) | p. (Glutamine422*) |
| PKD1 variant | — | — | c.11257C>T (p.Arg3753Trp) | — |
| Clinical Characteristics | | | | |
| Intellectual disability | Severe | Severe | Severe | Severe (IQ:30–39) |
| Renal cysts | — | USG: a 14-mm-diameter cortical cyst and a 10-mm-diameter septate cortical cyst were observed in the upper pole of the left kidney.<br><br>CT abdomen: 1. dense calcification was observed in the right adrenal gland location; 2. cystic lesions were detected in the upper pole of the left kidney.<br><br>DMSA renal scan: the right kidney is in normal localization, normal size, and smooth contours, and shows homogeneous activity within normal limits. The left kidney is smaller in size and smooth-shaped than the right, and has a homogeneous, slightly lower radiopharmaceutical uptake compared with the right. The contribution of the right kidney to total kidney functions was calculated as 63%, and the contribution of the left kidney was calculated as 37%. | USG: a few millimeter-sized simple cortical cysts were observed in the right kidney. There are several thin-walled, parapelvic/cortical-located anechoic cysts in the left kidney, the largest of which measures 15 mm in diameter in the middle part, some of which have calcified walls. | — |
| Obesity | + | — | — | — |
| Autism spectrum | Stereotypical movements, echolalia+ | + | — | + |
| Seizure disorder | — | — | Generalized tonic–clonic seizures | Generalized tonic–clonic seizures |
| Development | Expressive language difficulties, occasionally unresponsive to verbal stimuli at age 4. | Eye contact at 1.5 yr old. | Started walking at 4 yr old. Started using single words at 4 yr and forms non-coherent single- and two-word sentences. | Started walking at 2 yr old. Non-coherent words. |
| Other | Obesity (> 97 percentile). Height in 25–50 percentile. | Overweight, moderate persistent asthma, cafe-au-lait single submental lesion, hypermobility of joints. | Length: < 0.02 percentile.<br>Head circumference: < 5th percentile.<br>Dysmorphic facies: micrognathia, beaked nose, long philtrum, thin narrow mouth | Dysmorphic facies: beaked nose, prominent chin, thick eyebrows, deep-set eyes. Nail hypoplasia. |

(Fig 1G, yellow arrows). To analyze the impact of *cc2d1a* depletion on renal development, we used a two-cell injection strategy and depleted *cc2d1a* using the CRISPR/Cas9 system and quantified post-editing using the ICE (Inference of CRISPR Edits) algorithm (51) (Fig S1A and B). *Cc2d1a* CR#1+CAS9 +Dextran, Alexa Fluor 488 was injected into one of two cells of embryos at the two-cell stage. With this strategy, the uninjected side served as the internal control (Fig 1H). The embryos were then grown to st.45 (post-fertilization day 4 at 26°C) and visualized with optical coherence tomography (OCT) imaging, where we acquired the 3D scan of the entire tadpole in vivo (Fig 1I), as our group previously described (52). We measured the largest cross sections of the kidney for both sides. In comparison, the side of the embryo depleted of *cc2d1a* had overall larger, dysplastic kidneys than the control side (Fig 1I and J). Together with the observation of *cc2d1a* expression in the nephrostomes, and depletion of *cc2d1a* leading to kidney dysplasia, these findings closely recapitulated the kidney phenotype of our patients.

### *Cc2d1a* mutant fruit flies recapitulated the antisocial behavior

To determine how *CC2D1A* affects social behavior because the null and brain-specific knockout mice died shortly after birth, we used the *Drosophila* model to study social behavior (47). We used a well-established quantitative behavioral assay that tests *Drosophila* social interaction, one of the three core ASD phenotypes as defined by Diagnostic and Statistical Manual for Mental Disorders, fifth edition (DSM-5; American Psychiatric Association, 2013). One of the hallmarks of human ASD is the lack of proper interaction with other individuals, which includes inappropriate responses to social cues, causing them to either violate another person's "personal space" or overreact when another individual invades their personal space. Hence, we used the assay for fly social spacing (analogous to human social reciprocity), which exploits the natural tendency of flies where, when housed as a group, flies settle into a "comfortable" social spacing that can be quantified using the social space triangle (47).

Using the UAS-Gal4 system and actin-gal4 to drive the ubiquitous expression of l(2)gd1 RNAi (*l(2)gd1-IR*), we assessed the functional consequence of disrupting the *lethal (2) giant discs* (*Lgd*), the *Drosophila* ortholog of *CC2D1A*, on the control of social spacing behavior and the social space index (SSI) computed (47) (an SSI score of ≤0 suggests little or no social interaction). The social space was quantified for males, females, and males plus females, and we found a significant impact on the SSI in all three combinations as compared to isogenic controls supporting the role of *CC2D1A* in ASD (Fig S2A–C).

### *Cc2d1a* is required for proper left–right patterning

When we knocked down *cc2d1a* in *Xenopus* to examine its role during development, one of the striking impacts was the disruption of proper heart formation, often leading to the early demise of the tadpoles. A critical step during heart development is the looping process, when the tubular heart twists and loops around, forming the chambers of the heart. Under normal conditions, the tubular heart loops to the right, and this process relies on the proper LR signaling of the body axis. In *Xenopus*, the cardiac sac is transparent, allowing us to examine the cardiac looping and differentiate

a normally D-looped heart (dextra-looped—to the right) from abnormal phenotypes L-looped (levo-looped) or an A-looped (ambiguous-looped) heart, where the outflow tract twists to the left or has an indeterminate midline position (Fig 2A). When we knocked down *cc2d1a* and raised the embryos to st.46 (post-fertilization day 4 at 26°C), we noticed that hearts were abnormally looped (Fig 2B). Respectively, we used two non-overlapping CRISPRs targeting exon 1 (CRISPR#1) and exon 2 (CRISPR#2), which displayed 24% and 17% looping defects (Figs 2B and S1A and B). Because proper heart looping relies on proper LR patterning, we examined an upstream marker of LR asymmetry, homeobox gene *pitx2*, that emanates from the left lateral plate mesoderm in chick, mouse, and *Xenopus* (53, 54, 55, 56) (Fig 2C). When we knocked down *cc2d1a*, 26% (CR#1) and 22% (CR#2) of the embryos displayed the abnormal expression of *pitx2* at st.28 (post-fertilization day 3 at 26°C) as reversed, bilateral, or absent, suggesting that the lateral plate mesoderm did not correctly receive LR signaling (Fig 2D). We then examined the singling upstream to *pitx2* generated by the GRP.

The GRP is a ciliated structure that transiently forms at the dorsal layer of the blastula, analogous to the Kupffer's vesicle in zebrafish and the node in mice and humans, and is responsible for establishing LR body axis patterning (57) (Fig 2E). We first asked whether *cc2d1a* was expressed in the GRP, and using ISH, indeed, we revealed that it was (Fig 2E–G). We then examined the expression of the LR marker *dand5* to analyze the impact of *cc2d1a* knockdown on GRP patterning. During development, *dand5* is a nodal antagonist initially present symmetrically on the GRP at stages 14–16 (Fig 2H). Then, on the surface of the GRP, motile cilia emerge and start to beat at st.18–19, leading to a right-to-left fluid flow (gray arrows), causing *dand5* expression to be reduced on the left side of the embryo (Fig 2I). This asymmetric inhibition of *dand5* then activates a signaling cascade downstream that leads the transcription factor, *pitx2*, to be up-regulated on the left lateral plate mesoderm (Fig 2I), setting the proper left–right axis (58, 59, 60, 61). Interestingly, we observed the abnormal expression of *dand5* at post-flow st.18, with 32% (CR#1) and 32% (CR#2) of embryos having bilateral, reversed, or reduced/absent signal; however, we noticed a normal expression of *dand5* at pre-flow st.15 (Fig 2J). These findings suggested that *cc2d1a* depletion results in a reduction of the *dand5* inhibition on the left side of the embryo, hinting at the possibility that cilium-driven flow might have been compromised when *cc2d1a* is depleted. Before we further investigated the potential impact on GRP cilia, we asked whether the cardiac defect is specific to the *cc2d1a* knockdown via a rescue experiment.

### WT *CC2D1A* rescues abnormal LR patterning, and the patient variant is detrimental to protein function

Our data so far suggested that *cc2d1a* regulates left–right patterning; therefore, we examined cardiac looping in our rescue experiments. We injected CR#1 and Cas9 protein at the one-cell stage, followed by a co-injection with WT *CC2D1A* human RNA, and raised these tadpoles to stage 46 (day 4) to score for cardiac looping. The looping phenotype improved by 15% (Fig 2K). We, in parallel, also tested the co-injection of the GFP-tagged human *CC2D1A* RNA, which also rescued looping by 50%. However, introducing the patient variant RNA (c.1186C>T [p.Arg396*]) to *cc2d1a*-

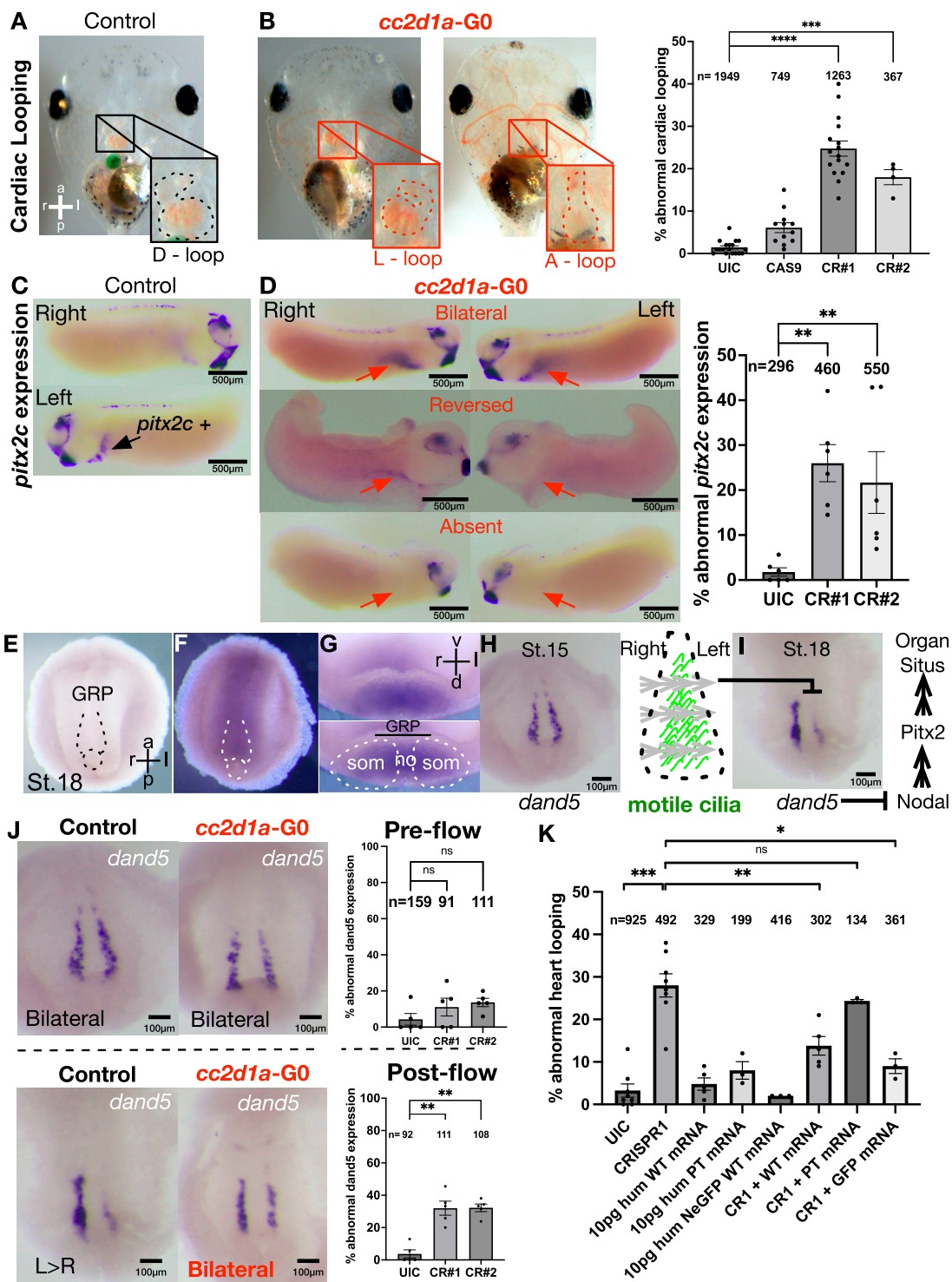

**Figure 2. cc2d1a is required for proper left–right patterning.**

**(A, B)** Representative images of cardiac outflow tract morphology and looping phenotypes in ventrally imaged st.45 *Xenopus* embryos. *cc2d1a* mutants with abnormal looping (L-loop and A-loop) are shown in red. Quantification of % cardiac looping defects in uninjected control (UIC) embryos, Cas9-only control, and *cc2d1a* CRISPR#1 and CRISPR#2. CR#1- and CR#2-injected embryos displayed 24% and 17% looping defects, respectively. Data are shown as the mean ± SEM. Black dots represent individual experiments, and the number of embryos is indicated above columns. P < 0.001 (***), P < 0.0001 (****). a, anterior; p, posterior; l, left; r, right; D-loop, dextra-looped; L-loop, levo-looped; A-looped, ambiguous-looped; G0, generation 0; UIC, uninjected control; CR, CRISPR. **(C)** Whole-mount in situ hybridization of st.28 control embryos showing the *pitx2c* signal (black arrow) on the left lateral plate mesoderm. **(D)** Representative images of st.28 *cc2d1a* mutant embryos with abnormal *pitx2c* expression patterns (bilateral, reversed expression, or absent signal). Red arrows indicate the location of the present or missing *pitx2c* signal. CR#1- and CR#2-injected embryos displayed 26% and 22% abnormal *pitx2c* expression, respectively. P < 0.01 (**). **(E, F)** Whole-mount in situ hybridization images of dissected GRPs from st.18 *Xenopus*

depleted embryos failed to restore proper heart looping (Fig 2K). Instead, the percentage of embryos co-injected with CR#1 and the patient variant RNA observed were close to 25% abnormal looping, suggesting that the patient variant is detrimental to function (Fig 2K). To further examine the role of *cc2d1a* in LR patterning, we turned to the GRP cilia.

### Cc2d1a is expressed at the GRP and localized to the base of the monocilia, and its depletion resulted in abnormal cilia

Briefly, cilia are evolutionary-conserved, centriole-derived, microtubule-based organelles protruding from the apical membrane of the cells. There are three types of cilia: immotile monocilia (sensory), motile monocilia, and motile multicilia. Given the results so far, *cc2d1a*-depleted embryos exhibited loss of *dand5* inhibition, which relies on the presence of motile monocilium-driven fluid flow. When we knocked down *cc2d1a* and analyzed the GRP cilia with IHC, we observed abnormal monocilia (Fig 3A and B). Anti-Arl13b antibody was used to label the monocilia, and phalloidin was used to label the actin to mark cell borders. Indeed, GRP cilia morphologically looked short and were depleted when *cc2d1a* was knocked down. The average size of the GRP was unchanged, yet the cilia per area were less (Fig 3C and D). When we injected N-GFP-*CC2D1A* RNA to investigate the localization, we found that *cc2d1a* was localized to the base of the GRP monocilia (Fig 3E and F). We then proceeded to determine whether *cc2d1a* ciliary localization and function are restricted to the GRP monocilia or are common to other cilium types. We first turned to the motile MCCs on the *Xenopus* epidermis.

### Cc2d1a is localized to the base of the epidermal multicilia, and its depletion leads to the loss of cilia; however, basal bodies were preserved

The epidermis of the *Xenopus* is populated with MCCs analogous to the human respiratory tract. We asked whether epidermal cilia are also regulated by *cc2d1a*. First, similar to our findings in the GRP, the N-GFP-*CC2D1A* localized to the base of the cilia on the MCCs. Next, we labeled the rootlets to better understand the localization in both GRP and epidermis. Rootlets are the cytoskeletal structure that originates from the centrioles and anchors the basal bodies of the cilia to the cell. We co-injected N-GFP-*CC2D1A* and CLAMP-RFP (calponin homology and microtubule-associated protein to label rootlets (62)) to visualize both proteins. *CC2D1A* was localized to the tip of the ciliary rootlets, showing the exact localization in both GRP

monocilia and epidermal multiciliated cells (Figs 3G–I and S3A–F). To determine whether the multicilia were abnormal like the GRP cilia, we again used two-cell injections to deplete *cc2d1a* on the one side. The embryos were grown to stages 28–30; then, we used OCT to visualize cilium-driven flow along the epidermis as previously described (63). Then, using IHC, we labeled the cilia with an anti-acetylated tubulin antibody. We confirmed the loss of epidermal cilia and cilium-driven fluid flow on the injected side with both modalities (Video 1—Fig S4A–C). To further explain this loss of ciliary phenotype, we investigated the centrioles at the base of the cilia, referred to as basal bodies. Basal bodies are modified centrioles that act as the microtubule organizer to form the cilia and are localized at the tip of the rootlets, identical to the *cc2d1a* localization. When we used IHC to co-stain the cilia with anti-acetylated tubulin and the basal bodies with anti-gamma tubulin in our *cc2d1a*-depleted tadpoles, we observed a loss of cilia but did not appreciate a loss or disorganization of the basal bodies, suggesting that despite defective ciliogenesis, centriole duplication, apicobasal migration, and proper membrane docking of the basal bodies remained intact (Fig S4D).

Given that *cc2d1a* depletion led to two discrete types of ciliary loss, GRP monocilia and epidermal multicilia, an intriguing question is how these findings might be relevant to the intellectual disability and autism spectrum disease that our patients and others in the literature are presenting with. For this reason, we examined the cilia in tadpole's central nervous system.

### *Cc2d1a* knockdown causes loss of cilium-driven CSF circulation in the midbrain

In the *Xenopus* tadpole brain, *cc2d1a* expression showed a specific location along the diencephalon and mesencephalon transition zone, encapsulating the cerebral aqueduct (Fig 4D and E). Interestingly, when we analyzed the cilium-driven CSF circulation, *cc2d1a* depletion led to the loss of CSF circulation in this specific region.

Ependymal cilium-driven CSF circulation can be visualized in *Xenopus* by OCT imaging. We have previously shown that the entire *Xenopus* brain ventricular system can be visualized, and CSF flow can be mapped using OCT imaging (64, 65). For this analysis, we obtained in vivo optical midsagittal cross sections of st.46 *Xenopus* brains to visualize brain morphology in WT and *cc2d1a* mutants (Fig 4A). The *Xenopus* brain has four ventricles: lateral ventricle (telencephalic), third ventricle (diencephalic), midbrain ventricle (mesencephalic), and fourth ventricle (rhombencephalic). We have

---

embryos with sense (E) and antisense (F) probes against *cc2d1a* RNA. The black and white dotted area delineates the GRP, a ciliated structure that helps establish left–right patterning. GRP, gastrocoel roof plate. **(G)** GRP cross section shows *cc2d1a* expression in the somites and notochord. GRP, gastrocoel roof plate; som, somites; no, notochord; v, ventral; d, dorsal. **(H)** At st.15, an upstream LR patterning marker, *dand5*, is present symmetrically on both sides of the GRP. Motile cilia (green lines) emerge and start to beat from right to left around st.18–19. The gray arrows indicate the fluid flow generated by the cilia. **(I)** Leftward fluid flow inhibits *dand5* expression on the left side of the GRP, leading to up-regulated *pitx2c*, which results in correct left–right asymmetry/organ situs. **(J)** Whole-mount in situ hybridization images of st.15 pre-flow GRPs (top panel) show bilateral *dand5* expression in *cc2d1a* mutant embryos, resembling the UICs. St.18 post-flow (bottom panel) *cc2d1a* mutant GRPs show bilateral *dand5* expression unlike the asymmetric expression shown in UICs. CR#1 and CR#2 GRPs both showed 32% abnormal post-flow *dand5* expression. ns, not significant; $P < 0.01$ (**). **(K)** Quantification of % abnormal heart looping in UICs, CR#1 knockdown, human WT *CC2D1A* mRNA (10 pg), *CC2D1A* c.1186C>T p.Arg396* human patient variant mRNA (10 pg), and human N-GFP WT *CC2D1A* mRNA (10 pg). *cc2d1a* mutant heart looping defects are rescued by human WT *CC2D1A* mRNA (15% rescue) and N-GFP WT *CC2D1A* mRNA (50% rescue). The rescue fails with the patient variant mRNA (25% abnormal looping). Data are shown as the mean ± SEM. ns, not significant; $P < 0.05$ (*), $P < 0.01$ (**), $P < 0.001$ (***). pg, picograms; WT, wild type; GFP, green fluorescent protein; hum, human.

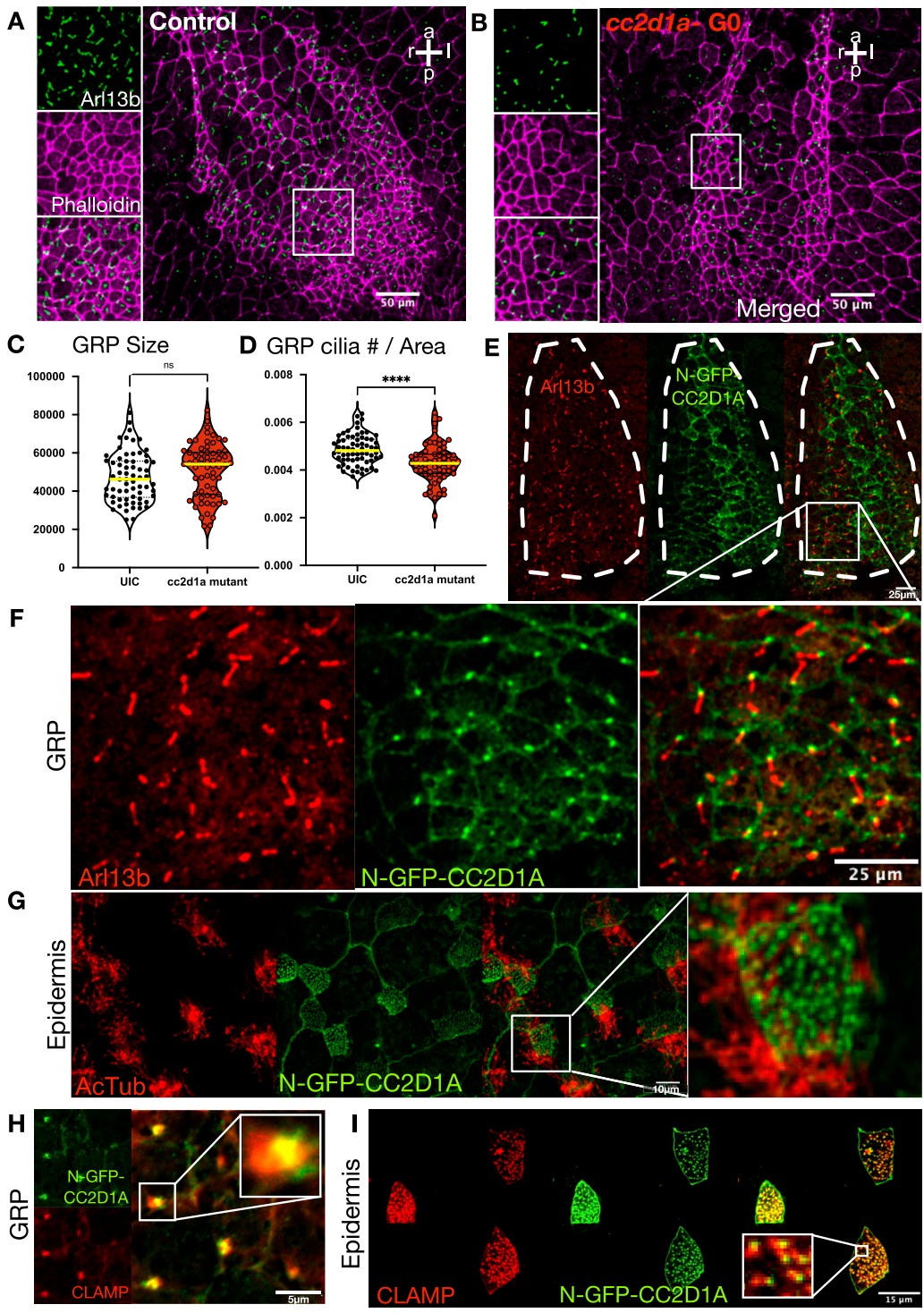

**Figure 3. CC2D1A is expressed at the GRP and localized to the base of the monocilia, and its depletion results in abnormal cilia.**
**(A, B)** Immunofluorescence images of control (A) and *cc2d1a*-depleted GRPs (B) stained with anti-Arl13b antibody (green) and merged with phalloidin (purple). Scale bar = 50 μm. **(C, D)** Quantification of the GRP size and (D) number of cilia per GRP area. Compared with the control, *cc2d1a* mutant embryos showed no significant difference in the GRP size, but did have fewer GRP cilia per area. ns, not significant; *P* < 0.0001 (****). **(E)** Immunofluorescence images of GRP (outlined in white) expressing human N-GFP CC2D1A (green) stained with anti-Arl13b (red). **(F)** Zoomed-in images of GRP. Human N-GFP CC2D1A is expressed in the base of the monocilia. Scale bar = 25 μm. **(G)** Immunofluorescence images of the epidermis of st.28–30 embryos expressing human N-GFP CC2D1A (green) stained with anti-AcTub (red) to show cilia. Human N-GFP-CC2D1A localizes to the base of cilia in multiciliated cells (MCCs). The rightmost merged image is a close-up of a MCC. Scale bar = 10 μm. **(H, I)** St.28–30 embryos expressing human N-GFP-CC2D1A (green) and CLAMP-RFP (red), which marks the rootlets. **(H, I)** Merged images show CC2D1A localizes to the tip of the ciliary rootlets in both GRP monocilia ((H); scale bar = 5 μm) and epidermal MCCs ((I); scale bar = 15 μm).

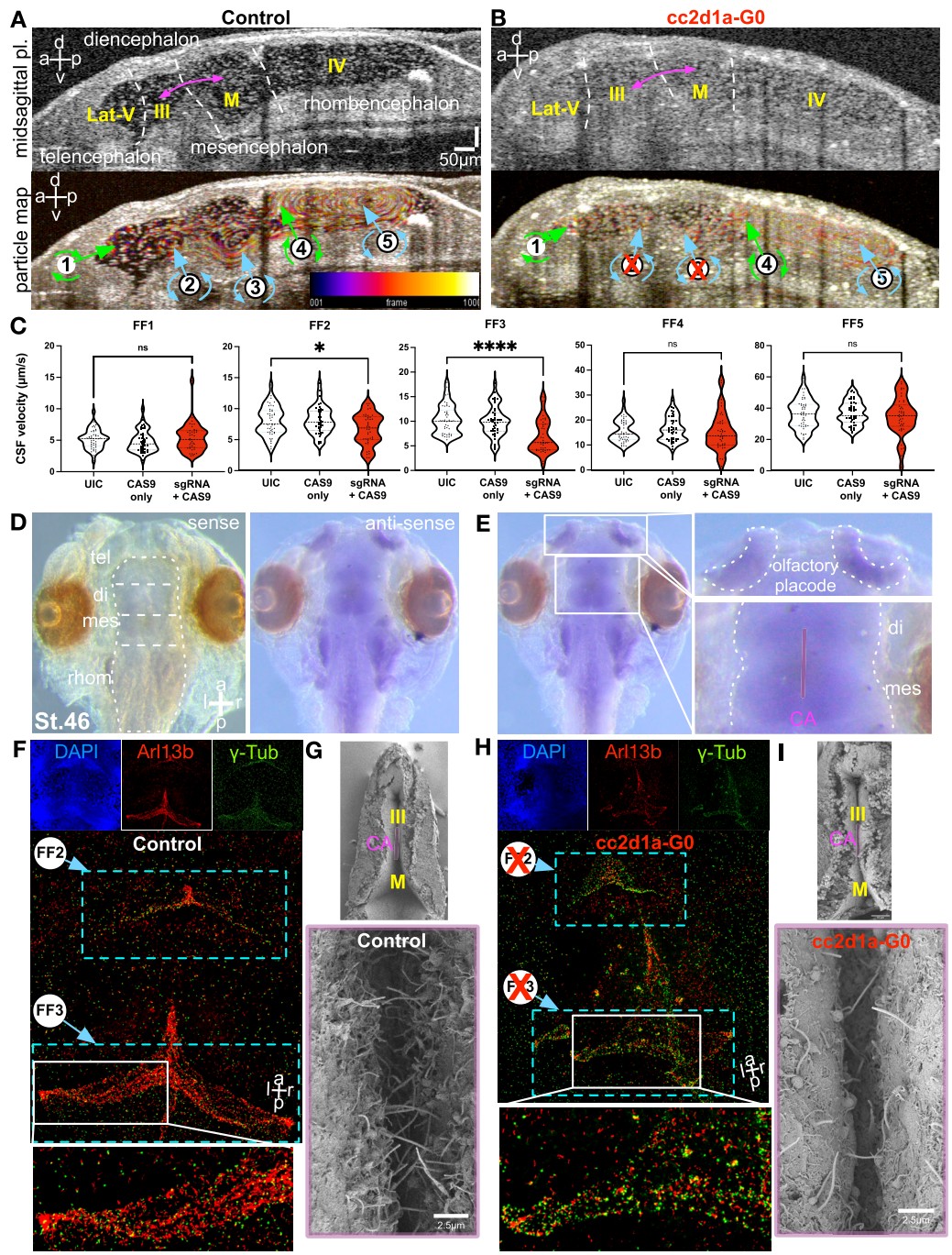

**Figure 4. CSF flow is regionally compromised in cc2d1a mutant embryo brains with defective cilia.**

**(A, B)** OCT-captured midsagittal section of st.46 control (A) and *cc2d1a* mutant (B) tadpole brains. White lettering and dotted white lines indicate brain regions; the brain ventricular system is labeled in yellow, and pink arrows indicate the location of the cerebral aqueduct. Ventricular CSF circulation was determined by particle tracking map in the control st.46 tadpole brain and the *cc2d1a* mutant tadpole brain. Green arrows indicate clockwise flow in the FF1 region and the FF4 region, whereas blue arrows indicate anti-clockwise fluid flow in FF2, FF3, and FF5. Compromised CSF flow in the cerebral aqueduct is indicated by the red X. Scale bar = 50 μm. Temporal color coding illustrates particle trajectory over time. The color scale bar shows the correspondence between the color and frame number in the color-coded image. a, anterior; p, posterior; d, dorsal; v, ventral; Lat-V, lateral ventricle; III, third ventricle; M, midbrain ventricle; IV, fourth ventricle. **(C)** Quantification of CSF flow velocities in UIC, Cas9 control, and *cc2d1a* mutant L-looped tadpoles, measured in μm/s, indicates slower FF2 and FF3 (cerebral aqueduct region) velocities in *cc2d1a* mutant L-looped tadpoles. ns, not significant; $P < 0.05$ (*), $P < 0.0001$ (****). CA, cerebral aqueduct; CSF, cerebrospinal fluid. **(D, E)** Whole-mount in situ hybridization on dissected WT *X. tropicalis* embryo brains at st.45 with antisense (right image) and sense (left image) probes against *cc2d1a* mRNA (D). **(E)** Cc2d1a is expressed in the diencephalon and mesencephalon transition zone, surrounding the cerebral aqueduct (E). a, anterior; p, posterior; d, dorsal; v, ventral; tel, telencephalon; di, diencephalon; mes, mesencephalon; rhom, rhombencephalon; CA, cerebral aqueduct; st., stage. **(F)** Immunofluorescence images of dissected st.46 control stained with DAPI (top left in blue), anti-Arl13b (red, top middle), and anti-γ-Tub (green, top right). The bottom shows merged images of Arl13b and γ-Tub. **(G)** SEM images of the ventral ependymal surfaces of st.46 control. **(H)** Immunofluorescence images of dissected st.46 *cc2d1a* mutant tadpole brains. In the *cc2d1a* mutant brains, cilia in the cerebral aqueduct region are compromised,

previously shown that each ventricle is decorated with ependymal motile cilia and generates local flow fields (FFs)—FF1–5 (65). These cilium-driven FFs have precise planar polarization and different velocities based on location. Here, we marked them clockwise versus anti-clockwise (Fig 4, Video 2). When we knocked down *cc2d1a*, we didn't observe a gross morphological change in the brain. However, the FF2 and FF3 showed severely diminished or near-absent CSF flow (Fig 4A–C and Video 2). The brain's third ventricle connects to the midbrain ventricle via the cerebral aqueduct (purple arrow, Fig 4A). Similar to the mammalian brain, aqueducts in *Xenopus* connect the ventricles and allow the transport of molecules between the ventricles essential for proper neurodevelopment. The CSF currents that regulate fluid transport along the cerebral aqueduct are lost when *cc2d1a* is depleted (Fig 4 and Video 2). The aqueduct where the CSF currents are lost is located in the diencephalon–mesencephalon transition zone where *cc2d1a* expression is enriched based on our ISH data (Fig 4). Based on these findings, we next examined the ciliary morphology of the ependymal surface. Flow fields 2 and 3 are localized to the ventral ependymal surface. We analyzed the ventral surface using IHC, where we marked cilia with anti-Arl13b and basal bodies with anti-gamma tubulin, and also used scanning electron microscopy for detailed analysis of the ciliary morphology. Both analyses showed severely disrupted cilia along the aqueduct (Fig 4G–I), explaining the loss of local CSF circulation in this region.

We marked cilia with anti-Arl13b and basal bodies with gamma tubulin, then analyzed the ventral surface using IHC. The analysis showed severely disrupted cilia along the aqueduct (Fig 4F–H), explaining the loss of local CSF circulation in this region.

*Cc2d1a* depletion led to defective cilia in *Xenopus's* GRP, epidermis, and ependyma. We finally asked whether patient-derived fibroblasts demonstrated any ciliary defects.

### Cultured fibroblast cells of patients demonstrate ciliary defects

Fibroblasts are mesenchymal cells of the connective tissue producing the extracellular matrix and collagen, involved in wound healing and scarring. When fibroblasts are cultured in low serum medium for ~48 h, they form cilia (66, 67), and multiple works demonstrated the in vitro use of fibroblasts to study ciliogenesis (68, 69, 70). To obtain fibroblasts, skin punch biopsies were taken from the patients and parents as described in the Materials and Methods section. Western blot analysis of fibroblast cell line lysates shows elimination of detectable CC2D1A expression in the two patient fibroblasts compared with the control fibroblasts and the father's fibroblasts (Fig 5A). To analyze the cilia in fibroblasts, we immunostained for acetylated tubulin to show cilia and phalloidin to show the cytoskeleton. Under normal conditions, control fibroblasts showed a monocilium with an average length of 3.50 ± 1.15 $\mu m$ (n = 590). In our index patients, either there were fewer ciliated cells (Fig 5B) or the ciliary length was significantly diminished (Fig 5C–F).

## Discussion

The CC2D1A gene has been identified in patients with a spectrum of neurodevelopmental diseases, including non-syndromic autosomal recessive intellectual disability, ASD, and seizures, as well as in patients with heterotaxy syndromes. Our work further expands the clinical presentation and reports patients with *CC2D1A* variants presenting with uni- and bilateral multicystic dysplastic kidney disease. Although we must acknowledge the confounding PKD1 variant in the 25-yo patient from Family 2 as a contributing cause of her cystic kidney disease, we present evidence that CC2D1A is also a contributing factor and could plausibly explain such a severe kidney phenotype at such an early age, as most patients with dominant PKD1 variants present as adults. Interestingly, our analysis of *cc2d1a* expression during early development revealed its association with nephrogenesis and the ciliated structures. *Cc2d1a* is specifically expressed at the three branches of the pronephric tubule (nephrostomes), known to be densely ciliated, but this expression pattern is not limited to the kidneys. *Cc2d1a* is highly expressed in the ciliated tissues throughout early embryonic development. The GRP, epidermis, nephrostomes, pronephric duct, optic vesicle, otic vesicle, olfactory placodes, and ciliated ependymal surface of the brain, specifically in the diencephalon and mesencephalon regions of the brain, showed significant *cc2d1a* expression. We also showed the *cc2d1a* localization at the base of mono- and multicilia in discretely different cell types, suggesting a potential global role in ciliogenesis. These findings align well with the recent work from reference 31 where the authors identified 26 probands with congenital heart disease, explicitly presenting with heterotaxy, which is well associated with ciliopathy. The authors also showed that TALEN-induced somatic *cc2d1a* knockdown in the zebrafish model recapitulated the patient heterotaxy phenotype and showed defective cilia in the central spinal canal (31). Of note, our brain findings in *Xenopus* are different than the zebrafish findings in reference 31 report, where we see defective cilia in the midbrain; however, the ependymal cilia in the hindbrain and spine remain unaffected.

Neurological defects are common in ciliopathies (71), and cilia are known to play a critical role in cerebral cortex development (72). Diverse neuropathologies in humans are associated with cilia, including Joubert, Bardet–Biedl, Meckel–Gruber, and orofaciodigital syndromes. It has been shown that cilia are involved in many processes in neurodevelopment, including progenitor regulation (73), interneuron migration (74), neural tube formation (75), and cerebellar development (76). Recent advances also demonstrated the role of cilia in embryonic CSF circulation (64, 65). This work highlights a local, specific loss of CSF circulation in the midbrain region when *cc2d1a* is lost, suggesting that cilia may also have additional roles in regional brain development and function. However, understanding the intricate relationship between local cilium-based CSF circulation, brain development, and human

leading to slower CSF flow in FF2 and FF3 (light blue boxes). **(I)** SEM images of the ventral ependymal surfaces of st.46 *cc2d1a* mutant tadpole brains in the CA region showing vastly reduced ciliary density. a, anterior; p, posterior; l, left; r, right; FF, flow field; CA, cerebral aqueduct; III, third ventricle; M, midbrain ventricle.

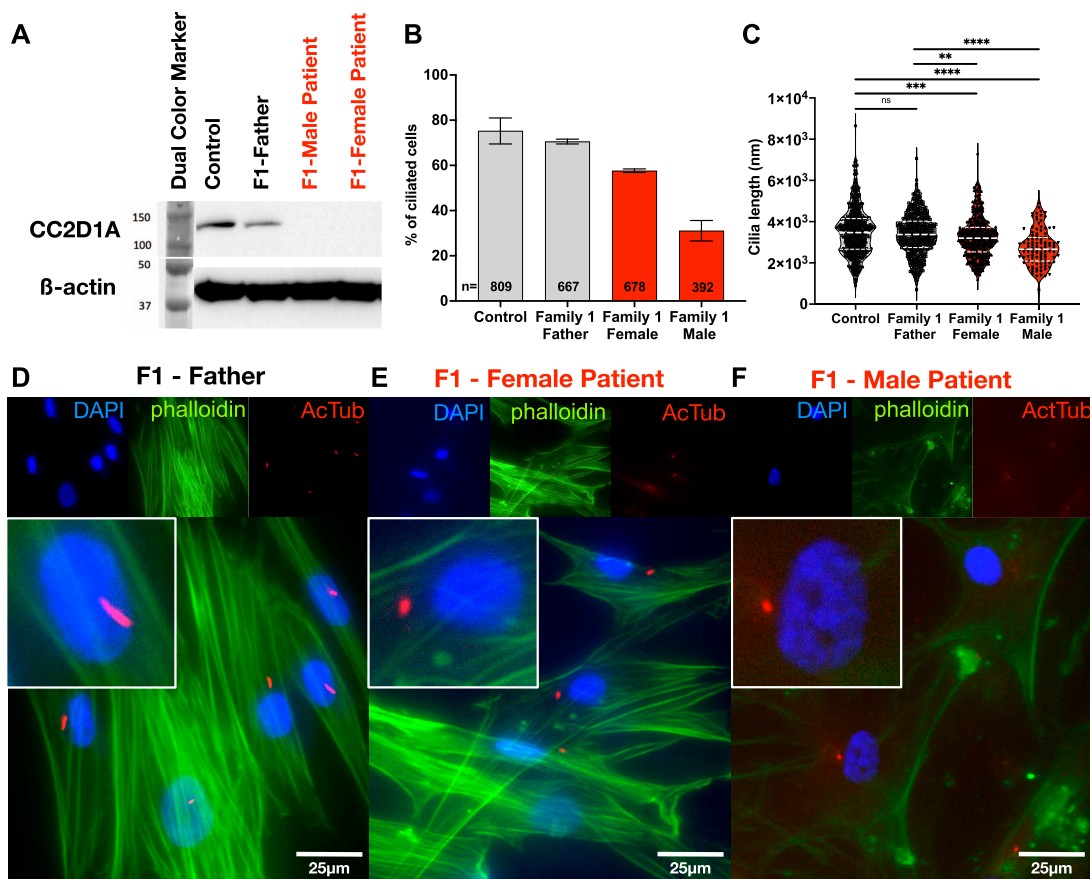

**Figure 5. Cultured fibroblast cells of patients demonstrate ciliary defects.**
**(A)** Western blot of fibroblast cell line lysates from control, and Family#1 father, female patient, and male patient. The affected patient cell lines did not show detectable CC2D1A expression. **(B, C)** Quantification of fibroblast starvation–induced ciliary length. The patients displayed fewer ciliated cells or shorter ciliary lengths. ns, not significant; $P < 0.01$ (**), $P < 0.0001$ (****). **(D, E, F)** Cultured and starved fibroblasts from Family#1 father, female patient, and male patient stained with DAPI (blue) to show nuclei, phalloidin, Alexa Fluor 488 (green) to show actin cytoskeleton, and anti-acetylated tubulin (red) to show ciliary axonemes. Cilia on the patient fibroblasts were less frequent and shorter than those on the control and father fibroblasts. Scale bar = 25 $\mu$m.
Source data are available for this figure.

neurocognitive disease remains incomplete. It will be important to understand this relationship to define the pathophysiology better.

In summary, we have added evidence for CC2D1A as a cause of a multisystem ciliopathy syndrome, expanding the spectrum of CC2D1A-related disease from neurodevelopmental and cardiac involvement to the currently included cystic renal disease, LR patterning and CSF circulation. We also provide functional support for novel variants associated with this emerging disease and introduce new understandings of ciliary biology as relates to CC2D1A.

# Materials and Methods

## DNA sequencing and bioinformatics processing

### Family 1
Written informed consent for participation in the study was gained from both parents. Genomic DNA was extracted from

peripheral blood samples of patients and their parents using QIAamp DNA Mini Kit (QIAGEN). The Clinical Exome Solution (SOPHiA GENETICS) was used for exome enrichment. All procedures were carried out according to the manufacturer's protocols. It is a capture-based target enrichment kit and covers 4,900 genes with known inherited disease-causing mutations. Paired-end sequencing was performed on an Illumina NextSeq 500 system in Bursa Uludag University with a read length of 150 × 2. Base calling and image analysis were conducted using Illumina's Real-Time Analysis software. The BCL (base calls) binary is converted into FASTQ using the Illumina package bcl2fastq.

### Bioinformatics analysis
All bioinformatics analysis was performed on the SOPHiA DDM platform, which includes algorithms for alignment, calling SNPs and small indels (Pepper), calling copy-number variations (Muskat), and functional annotation (Moka). Raw reads were aligned to the human reference genome (GRCh37/hg19). Variant filtering and interpretation were performed on SOPHiA DDM. Raw data were

analyzed via the SOPHiA DDM data analysis platform. Alignment and variant discovery were performed by Pepper, a proprietary baseline algorithm from SOPHiA GENETICS. Variant annotation was performed with SOPHiA GENETICS' Moka software, and for each variant, the effect of the variant on the protein sequence (missense, stop gain, etc.), the frequency of occurrence in various populations (1000G, ESP, ExAC, gnomAD), and prediction algorithms (SIFT, Poly-Phen) were determined. Information such as the destructive effect of the variant has been added. CNV detection was performed with SOPHiA GENETICS' Muskat software. Only variants located within exonic regions and the 20-base pair border region between exons and introns were included. Variants that passed the upstream pipeline filters and with a call quality of ≥20 were included. Variants with an allele frequency of >1% in GnomAD, 1000 Genomes, or ExAC were excluded. Homozygosity mapping was carried out in families with consanguineous marriages with HomSI (77).

### Family#2
Written informed consent for participation in the study was gained from both parents. Genomic DNA was extracted from peripheral blood samples of patients and their parents using QIAamp DNA Mini Kit (QIAGEN). Whole-exome sequencing analysis was performed using the xGen Exome Research Panel v2 kit through the next-generation sequencing method.

### Bioinformatics analysis
The resulting Variant Call Format data were analyzed by creating the following filter using QIAGEN Clinical Insight Interpret 8.1.20220121. Only variants located within exonic regions and the 20-base pair border region between exons and introns were included. Variants that passed the upstream pipeline filters and with a call quality of ≥20 were included. Variants with an allele frequency of >1% in GnomAD, 1000 Genomes, or ExAC were excluded.

### Xenopus husbandry

X. tropicalis were housed and cared for in our aquatics facility according to established protocols approved by the Yale Institutional Animal Care and Use Committee. Animals were housed at our aquatic facility under environmental control, including water temperature, pH, and conductivity, as the stability of these variables is essential. We followed the established protocol describing conditions to optimally raise and maintain X. tropicalis from embryos to adulthood (78).

### Generation of the *cc2d1a*, *pitx2*, and *dand5* probe and mRNA for whole-mount in situ hybridization

The plasmids were linearized using the listed restriction enzyme (SphI #R3182, Hind III #R3104, ClaI #R0197; NEB); then, the antisense RNA probes were produced using HiScribe T7 High Yield RNA Synthesis Kit (#E2040S; NEB) and DIG-dUTP (#03359247910; Roche).

### Whole-mount in situ hybridization

In situ hybridization was performed on fixed embryos following the standard protocol (80). However, the final fixation was done with 4% PFA (#158127; Sigma-Aldrich) and 0.1% glutaraldehyde (#5882; Sigma-Aldrich) in PBS (#P7059; Sigma-Aldrich) instead of Bouin's fixative. Expression patterns for *pitx2c* and *dand5* were assayed in stage 30 and stage 18 *cc2d1a* knockdown embryos, respectively, and *cc2d1a* expression pattern was also assayed in the WT stage 18 (GRP), stage 30 (epidermis, optic vesicle, otic vesicle, nephrostomes, pronephric duct), and stage 45 (craniofacial structures). Removal of pigment by incubation in bleaching solution (1% hydrogen peroxide [#H1009; Sigma-Aldrich] and 5% formamide [#F7503; Sigma-Aldrich] in 0.5x SSC [#AB13156; American Bioanalytical]) was done after rehydration, before the 5-min 0.01 mg/ml proteinase K (#AB00925; American Bioanalytical) treatment for stage 45 embryos. For all other stages, bleaching was done after the final fixation after the BM Purple (#11442074001; Sigma-Aldrich) color reaction.

### CRISPRs, mRNA, and injections

CRISPR sgRNAs (small guide RNAs) for *cc2d1a* were designed from the v9.0 model of the *X. tropicalis* genome using CRISPRscan (81):
    CRISPR-1 (exon 1): 5′-GGTCGGAAAGAAGTCCGTGGGGG-3′.
    CRISPR-2 (exon 15): 5′-GCGCTGTTGTTTGGAGCGAAGGG-3′.
    sgRNAs were produced using EnGen sgRNA Synthesis Kit (#E3322V; NEB). A pDONR221 plasmid containing a human *CC2D1A* insert (reference sequence NM_017721) was obtained from DNASU (#HsCD00829388). The insert was cloned into pCS DEST (#22423; Addgene) and 223 pCS EGFP DEST (#13071; Addgene) using Gateway cloning techniques (Invitrogen). The patient variant (c.1186C>T) was produced using Q5 Site-Directed Mutagenesis Kit (#E0554S; NEB) and also cloned into a pCS DEST vector. mRNA was produced from these plasmids using mMESSAGE mMACHINE SP6 Transcription Kit (#AM1340; Invitrogen). *Xenopus* embryos were produced by in vitro fertilization and raised to appropriate developmental stages in 1/9x MR + 50 µg/ml gentamicin (#G3632; Sigma-Aldrich) (82). Post-fertilization embryos were injected at the one-cell or two-cell stage according to standard protocols (82, 83, 84). 400 pg of CRISPR sgRNA combined with 1.6 ng Cas9 protein (#CP03; PNA Bio) and a fluorescent tracer, Dextran, Alexa Fluor 488 (#D22910; Invitrogen), was injected at a volume of 2 nl into each embryo at the one-cell

**Digoxigenin-labeled antisense mRNA probes against *cc2d1a*, *pitx2* (58, 79), and *dand5* (58, 79) were produced from the following plasmids.**

| Gene | Sequence reference | Expression vector | Linearizing restriction enzyme |
|------|--------------------|--------------------|-----------------|
| *cc2d1a* | BC161089 | pCS DEST | Sph I |
| *pitx2c* | TNeu083k20 | pCS 107 | Hind III |
| *dand5* | TEgg007d24 | pCS 107 | Cla I |

stage. For targeted loss-of-function experiments, 200 pg sgRNA with 0.8 ng Cas9 protein was injected into one of two cells at the two-cell stage; then, embryos were raised to the desired stages (85, 86). For rescue experiments, 10 pg of human WT *CC2D1A* mRNA, patient variant *CC2D1A* mRNA, or GFP-tagged *CC2D1A* mRNA was injected separately into knockdown embryos at the one-cell stage.

### Genotyping/CRISPR analysis

To extract and purify the genomic DNA, stage 45 CRISPR–injected embryos were dissociated in 50 µl of 50 mM NaOH (#7708-10; Macron) at 95°C for 10 min (flicked every 3 min). Samples were vortexed, then neutralized with 20 µl 1 M Tris (pH 7.4), and centrifuged for 5 min. Supernatants were collected and stored at −20°C. We designed primers around each CRISPR site for PCR amplification using Primer3plus and then performed PCR using Phusion High-Fidelity DNA Polymerase (#M0530S; NEB). The primers used were CRISPR-1 (exon 1): 5′-GAGCCCCCTGCATATAACCC-3′, 5′-GGGCACTGCTATTCTAGTTGC-3′, and CRISPR-2 (exon 15): 5′-CCTGGGACCTATTGCAAAGC-3′, 5′-CATCAGCACAGGAGCAAAGC-3′. The PCR products were run on an agarose gel; the fragments were gel-extracted and purified using Monarch DNA Gel Extraction Kit (#T1020S; NEB), then sent for Sanger sequencing (Quintarabio). Sequencing results were then analyzed for cutting efficiency using the Synthego ICE online tool. We verified that CRISPR/Cas9 edited the proper cut site (Fig S1). Genotyping results to confirm knockdown of the *cc2d1a* gene show a knockout score of up to 90% for CRISPR#1 (targeting exon 1) and a knockout score of up to 81% for CRISPR#2 (targeting exon 15). Overall knockdown over multiple samples is represented in Fig S1.

### *Drosophila* stocks and husbandry

*Drosophila* stocks were raised in standard food cornmeal/molasses/agar bottles or vials at 25°C with a relative humidity of 20–40% in a 12-h dark–light cycle. *l(2)gd1-IR*, isogenic control, and *Actin-gal4* flies were obtained from the Bloomington Drosophila Stock Center. All behavioral experiments were performed in a genotype-balanced manner. To minimize the disruption of standard environmental conditions, flies were reared in bottles and thus socially enriched, kept as mixed genders to allow mating, and kept with standard food at all times before testing. Flies were separated by gender the day before each experiment.

All experiments used flies naive to the test performed. Unless otherwise noted in the text, the flies were collected from the bottles when ~3–4 d old, sexed the day before the experiment, and placed in vials (40/vial). Experiments were performed at the same time of day from 11 am to 4 pm (between ZT4 and ZT9) to reduce variations between trials. All behavioral assays were performed with a white background using cardboard poster board and in a room at ~25°C and ambient light.

### *Drosophila* social space assay

The vertical triangle test chamber was constructed at the Yale Machine Shop using the dimensions described by Simon, et al.

Briefly, vertical triangle test chambers were assembled using two square glass plates (18 × 18 cm), separated by 0.47-cm spacers to restrict flies within a 2D space (47, 87). Four spacers were used and arranged into an isosceles triangle, with a 10-cm ruler placed on the top-right surface for analysis scaling. Vials from the incubator were allowed to acclimatize for 2 h before the experiment. The experiment was conducted over 2 d under specific gender conditions: male, female, and mixed (20♂/20♀), each with three independent repeats. Flies were transferred from the vials to the chambers, and after closing the entrance and securing the chambers with binder clips, the chamber was tapped uniformly three times on a smooth surface to standardize the starting position of the flies. Digital images were captured every 30 min, three times per trial. Post-trial, the flies were collected using the $CO_2$ diffuser and returned to their original vials for overnight recovery. This procedure was repeated the next day at the same starting time for consistency. In total, 12 social behavioral assays were performed for each condition over the 2 d.

Digital images were scaled using the in-frame ruler and converted into Tagged Image File Format files using ImageJ, followed by a batch conversion using Imaris software. Each fly was individually identified using the spot selection tool, excluding any appearing deceased. K-nearest neighbor distance was computed using Spots (Imaris). Image-specific data were then exported to Excel and Prism 9 for further analysis, categorizing values into 0.5-cm bins. The distribution of fly distances was visualized using binned histograms.

The SSI was derived using the binned distance histograms. The SSI was computed using the difference between the percentage of flies in the first bin and the percentage of flies in the second bin. An SSI below 0 indicated minimal to no social interaction among the flies. Non-parametric tests, including the Kolmogorov–Smirnov and Mann–Whitney tests, were applied to analyze the binned histograms and SSI.

### *Xenopus* cardiac looping

Post-fertilization stage 45 embryos were anesthetized in 2 g/liter Syncaine (tricaine methanesulfonate; Syndel) in 1/9x MR and ventrally scored for heart looping. The direction of heart looping was determined by the position of the outflow tract (D-loop if outflow tract curves to the right, L-loop if outflow tract loops to the left, and A-loop if it does not loop).

### IHC and imaging

#### *GRP monocilia (st.18)*
Control and *cc2d1a* knockdown embryos were raised to stage 18, fixed with 4% PFA in PBS for 1 h at RT, and rinsed with PBS. GRPs were dissected and incubated in a blocking buffer (3% BSA [#A9647; Sigma-Aldrich] and 0.1% Triton X-100 [#AB02025 in PBS; American Bioanalytical]) for 1 h at RT. Samples were then incubated with mouse anti-Arl13b (NeuroMab clone N295B/66) diluted 1:100 in blocking buffer overnight at 4°C. GRPs were washed with 0.1% Triton X-100 in PBS three times for 10 min, incubated in blocking buffer for 30 min, and then incubated in donkey anti-mouse Alexa Fluor 594 (#A21203; Invitrogen) diluted 1:500 in blocking buffer at RT for 2 h.

The samples were then washed with 0.1% Triton X-100 in PBS two times for 10 min at RT. Actin filaments were stained with phalloidin, Alexa Fluor 488 (#A12379; Invitrogen) diluted 1:100 in 0.1% Triton X-100 in PBS at RT for 1 h; then, the samples were washed in PBS two times for 10 min and mounted between coverslips with ProLong Gold antifade mountant (#P36934; Thermo Fisher Scientific).

### Epidermal multicilia (st.28–30)

The *Xenopus* epidermis is populated with multiciliated cells allowing straightforward observation and functional analyses (88, 89, 90). Control and *cc2d1a* knockdown embryos were raised to stages 28–30, then fixed and immunostained the same way as the GRPs instead of using mouse anti-acetylated tubulin clone 6-11B-1 (#T6793; Sigma-Aldrich) and rabbit anti-γ-tubulin (#T3559; Sigma-Aldrich) as the primary antibodies. Embryos were mounted between coverslips in ProLong Gold using vacuum grease as a spacer.

### Ependymal monocilia/multicilia (st.45)

Control and *cc2d1a* knockdown tadpoles were raised to stage 45. Mutant tadpoles were scored for heart looping defects; then, normal control and abnormally looped tadpoles were fixed with 4% PFA in PBS for 1 h and rinsed with PBS. To be able to better observe brains, embryo heads were dissected, removing facial cartilage, tail, gut, and lower jaw. The heads were then dehydrated by washing twice with 100% methanol (#179337; Sigma-Aldrich) and stored at −20°C overnight. The samples were then bleached in 10% hydrogen peroxide in 100% methanol at RT on direct light until the pigment was sufficiently gone (about 3 h). After a rinse in 100% methanol, the samples were rehydrated stepwise (50% methanol and 25% methanol, 10 min each) to TBS (155 mM NaCl [#9888; Sigma-Aldrich] and 10 mM Tris [#AB02000; American Bioanalytical], pH 7.5), then incubated in 0.1% Triton X-100 in TBS overnight at 4°C. The next day, embryos were blocked in 10% FBS (Sigma-Aldrich) and 0.3% Triton X-100 in BSDSGS (1% BSA, 5% donkey serum [#017-000-121; Jackson ImmunoResearch], 5% goat serum [#5425S; Cell Signaling Technology], 0.1% glycine [#AB00730-01000; American Bioanalytical], 0.1% lysine [#AB145111; Abcam] in PBS) for 4 h at RT, then incubated in primary antibodies overnight at 4°C as described previously (89). The primary antibodies used were rabbit anti-Arl13b (#17711-1-AP; Proteintech) diluted 1:100 and mouse anti-γ-tubulin (#T6557; Sigma-Aldrich) diluted 1:200 in 10% FBS, 0.1% Triton X-100 in BSDSGS. The embryos were rinsed, then washed in TBST (TBS + 0.1% Triton X-100) at RT for 1 h, washed in TBS three times for 1 h at RT, and washed in TBS overnight at 4°C. The next day, they were incubated in secondary antibodies, donkey anti-rabbit Alexa Fluor 594 and chicken anti-mouse Alexa Fluor 488 (#A21200; Invitrogen) each diluted 1:500; and Hoechst 33342 (#H3570; Invitrogen) diluted 1:5,000 in TBS for 2 h at RT. The samples were then washed in TBS three times for 10 min. Samples were then mounted between coverslips in Pro-Long Gold using vacuum grease as a spacer.

### Cultured fibroblasts

A 4-mm punch skin biopsy was taken under local anesthesia from the patients (genotype confirmed) and an age-matched, unrelated, healthy, donor control. Written informed consent for participation in the study was gained from the patients and control.

Sterile scalpel blades were used to cut the biopsies into smaller pieces of 0.5 mm, which were then put into a six-well plate. The tissue was incubated at 37°C in 5% CO2 in a humidified incubator with a limited amount of growth media (DMEM, 10% FBS, 1 mM sodium pyruvate, 4 mM L-glutamine, penicillin–streptomycin, and 2.5 μg/ml amphotericin B). Fresh medium (2 ml) was added the next day. The fibroblasts were cultured for ~4 wk until enough fibroblast outgrowth had occurred to allow for additional cell passage (amphotericin B [J67049.AD; Thermo Fisher Scientific], DMEM [#21068028; Gibco], L-glutamine [#25030081; Gibco], sodium pyruvate [#11360070; Gibco], penicillin–streptomycin [#10378016; Gibco], FBS [#10270-106; Gibco]). Passaging was done with 0.25% trypsin–EDTA (#25200056; Gibco). Ciliary growth was induced by incubating cultured cells for 48 h in growth medium without FBS. Cells to be immunostained were cultured in eight-well culture slides (#354108; Falcon/Corning), fixed in 4% PFA in PBS for 1 h, and then washed three times with PBS. Cells were then incubated in blocking buffer (3% BSA/PBS with 0.1% Triton X-100) for 1 h, then incubated overnight at 4°C in a primary antibody (mouse anti-acetylated tubulin; Sigma-Aldrich) diluted 1:1,000 in blocking buffer. Cells were washed three times for 10 min with PBS, then incubated in a secondary antibody (goat anti-mouse Texas Red, #T6390; Invitrogen) diluted 1:500 in blocking buffer. Wells were washed two times for 10 min with PBS, then incubated in phalloidin, Alexa Fluor 488 diluted 1:100 and Hoechst 33342 diluted 1:5,000 in PBS for 30 min. Wells were then washed two times for 10 min with PBS. Then, wells were removed from the culture slide and the cells were mounted with ProLong Gold.

All immunostained samples were imaged on a Zeiss LSM 880 confocal microscope. Fluorescent images were processed and analyzed using Fiji/ImageJ (91). For GRP ciliary quantification, the GRP area was outlined, and then, cilia were manually counted in that region as described previously (79, 92). Fibroblast ciliary lengths were measured using Zen (Blue Edition) version 3.6 software (Zeiss).

### Western blotting

To extract lysates from fibroblasts, cells were cultured in six-well culture plates to ~80% confluence, then washed once with PBS, and aspirated. SDS lysis buffer (2% SDS [#AB01922; American Bioanalytical], 10% glycerol [#2136-03; Baker], and 62.5 mM Tris, pH 6.8) was heated to 100°C; then, 150 μl was added to each well of the six-well plate. The cells were lysed by swirling the slurry on the bottom of the well with a pipet tip; then, the slurry was transferred to microfuge tubes. The samples were incubated at 100°C for 10 min, cooled to RT, and then stored at −20°C until needed. Samples were thawed on ice; then, protein concentrations were calculated using BCA Protein Assay Kit (#23225; Thermo Fisher Scientific) according to the manufacturer's instructions. 5 μg of each lysate with Laemmli sample buffer (#161-0747; Bio-Rad) was run on a 4–12% Bolt Bis-Tris Plus gel (#NW04120BOX; Invitrogen), then transferred to a PVDF membrane (#1620219; Bio-Rad). The membrane was blocked for 1 h with 5% non-fat dry milk (#AB10109; American Bioanalytical) in TBST, then incubated overnight at 4°C in primary antibody (mouse anti-CC2D1A, #H00054862-B01P; Thermo Fisher Scientific)

diluted 1:250 in 5% non-fat dry milk in TBST. The membrane was washed three times for 15 min in TBST, then incubated for 2 h in a secondary antibody (donkey anti-mouse HRP, #715-035-150; Jackson ImmunoResearch). The membrane was then washed three times for 15 min in TBST. The SuperSignal West Pico Plus chemiluminescent substrate (#34580; Thermo Fisher Scientific) was used according to the manufacturer's instructions to visualize stained protein bands. After exposure to the substrate, the membrane was scanned using an Azure c300 Western blot imager.

### OCT imaging and CSF velocity quantification

OCT imaging was performed as we previously demonstrated (52, 63, 64, 65, 93). Stage 46 tadpoles were anesthetized in 2 g/liter Syncaine in 1/9x MR, and cross-sectional (midsagittal) images of the brain ventricles were obtained with OCT/ThorImage. 2D/3D images and 2D Videos were used to quantify the brain areas and CSF flow velocities using Fiji/ImageJ and MATLAB. The Gaussian process post-processing was applied for particle velocimetry to quantify CSF flow velocity, as we described previously (63). CSF flow was measured in $\mu$m/sec. Figure images were built with averaged particle speed colorization and were processed in Fiji, ImageJ (91). CSF circulation flow was classified as normal, slow, or absent because of flow.

### Scanning electron microscopy

The dissected tissue was fixed with 4% PFA overnight at 40°C, followed by further fixation once the sample was pinned open with 2.5% glutaraldehyde in 0.1 M sodium cacodylate buffer, pH 7.4 (#16537-20; Electron Microscopy Sciences), for 1 h. Samples were rinsed in 0.1 M sodium cacodylate buffer (#J60344.AE; Thermo Fisher Scientific) and post-fixed in 2% osmium tetroxide (#201030; Sigma-Aldrich) in 0.1 M sodium cacodylate buffer, pH 7.4. These samples were rinsed in buffer and dehydrated through an ethanol series to 100%. The samples were dried using a Leica 300 critical point dryer with liquid carbon dioxide as transitional fluid and were glued to aluminum stubs using a carbon adhesive, and then sputter-coated with 4 nm platinum 80/palladium 20 using Cressington 208HR. The samples were viewed and digital images acquired in Zeiss CrossBeam 550 between 1.5 and 2 kV at a working distance of 8–12 mm.

### Statistical analysis

All experiments had at least three replicates and were tested for statistical significance using two-tailed $t$ tests in GraphPad Prism 9. Statistical significance was defined as $P < 0.05$ (*), $P < 0.01$ (**), $P < 0.001$ (***), and $P < 0.0001$ (****).

## Appendix: MarmaRare Group

Yasemin Alanay, Yasemin Kendir-Demirkol, Ozlem Akgun Dogan, Mahmut Cerkez, Ergoren, Ozden Hatirnaz Ng, Ugur Ozbek, Ozkan Ozdemir, Sebnem Ozemri Sag, Ilayda Sahin, Sehime G Temel, Kanay Yararbas.

## Data Availability

De-identified data are available upon request from the authors.

### Ethics declaration

All institutions involved in this research received approval from their local IRB or Research Ethics Committee. Informed consent was obtained from all individuals or their parents/legal guardians through the IRB protocols at Yale University School of Medicine (main IRB) or one participating institution. Individual data have been de-identified; for the presentation of identifiable patient images, express written consent has been obtained from the individuals or their parents/legal guardians. Animal research was performed under an approved Institutional Animal Care and Use Committee Protocol at the Yale University School of Medicine.

## Supplementary Information

## Acknowledgements

The authors thank all the patients and their families for participating in our research study. The authors thank the Yale Center for Genome Analysis for DNA sequencing, and Xinran Liu and Morven Graham at the Yale Electron Microscopy laboratory for assistance with micrographs. E Deniz was supported by NIH/NICHD R01NS127879.

### Author Contributions

AH Kim: conceptualization, formal analysis, investigation, visualization, methodology, and writing—original draft, review, and editing.
I Sakin: conceptualization, formal analysis, investigation, and writing—original draft, review, and editing.
S Viviano: conceptualization, resources, data curation, formal analysis, validation, investigation, visualization, methodology, and writing—original draft, review, and editing.
G Tuncel: data curation, formal analysis, validation, investigation, and methodology.
SM Aguilera: investigation, visualization, and methodology.
G Goles: investigation, visualization, and methodology.
L Jeffries: data curation, formal analysis, and writing—review and editing.
W Ji: data curation, formal analysis, validation, and writing—review and editing.
SA Lakhani: data curation and formal analysis.
CC Kose: data curation and investigation.
F Silan: data curation, formal analysis, and investigation.

SS Oner: investigation and writing—review and editing.

OI Kaplan: investigation and writing—review and editing.

MC Ergoren: data curation, formal analysis, investigation, methodology, and writing—original draft, review, and editing.

K Mishra-Gorur: data curation, formal analysis, supervision, investigation, methodology, project administration, and writing—original draft, review, and editing.

M Gunel: data curation, formal analysis, supervision, investigation, and project administration.

SO Sag: conceptualization, resources, data curation, formal analysis, supervision, validation, investigation, methodology, and writing—original draft, review, and editing.

SG Temel: conceptualization, resources, data curation, formal analysis, supervision, funding acquisition, validation, investigation, visualization, methodology, project administration, and writing—original draft, review, and editing.

E Deniz: conceptualization, resources, data curation, formal analysis, supervision, funding acquisition, validation, investigation, visualization, methodology, project administration, and writing—original draft, review, and editing.

## Conflict of Interest Statement

One author reports part ownership of startup companies unrelated to this work: Qiyas Higher Health (SA Lakhani) and Victory Genomics (SA Lakhani). All other authors declare no conflicts of interest.

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
