## [Reviewer comments · Life Science Alliance]

Life Science Alliance

CC2D1A causes ciliopathy, intellectual disability, heterotaxy, renal dysplasia and abnormal CSF flow

Angelina Kim, Irmak Sakin, Stephen Viviano, Gulden Tuncel, Stephanie Aguilera, Gizem Goles, Lauren Jeffries, Weizhen Ji, Saquib Lakhani, Canan Kose, Fatma Silan, Sukru Oner, Oktay Kaplan, MarmaRare Group, Mahmut Ergoren, Ketu Mishra-Gorur, Murat Gunel, Sebnem Sag, Sehime Temel, and Engin Deniz

DOI: <https://doi.org/10.26508/lsa.202402708>

Corresponding author(s): Engin Deniz, Yale School of Medicine and Sehime Temel, Uludağ University

Review Timeline:

Submission Date:	2024-03-11
Editorial Decision:	2024-04-19
Revision Received:	2024-07-18
Editorial Decision:	2024-07-19
Revision Received:	2024-07-29
Accepted:	2024-07-30

Transaction Report:

April 19, 2024

Re: Life Science Alliance manuscript #LSA-2024-02708

Dr. Engin Deniz
Yale School of Medicine
Pediatric Critical Care
333 Cedar st
New Haven 6520

Dear Dr. Deniz,

Thank you for submitting your manuscript entitled "CC2D1A causes ciliopathy, intellectual disability, heterotaxy, renal dysplasia and abnormal CSF flow" to Life Science Alliance. The manuscript was assessed by expert reviewers, whose comments are appended to this letter. We invite you to submit a revised manuscript addressing the Reviewer comments.

Thank you for this interesting contribution to Life Science Alliance. We are looking forward to receiving your revised manuscript.

Sincerely,

B. MANUSCRIPT ORGANIZATION AND FORMATTING:

Reviewer #1 (Comments to the Authors (Required)):

Kim et al., have demonstrated that cc2d1a deletion induces various ciliopathy phenotypes using *Xenopus*. The authors have identified two de novo mutations of cc2d1a in human. The authors used CRISPR/Cas9 system to delete cc2d1a and showed that cc2d1a deletion induced enlarged kidney, disrupted left right patterning and disrupted circulation of central spinal fluid. Finally, the authors showed that cc2d1a is located at the rootlet of the basal body. The paper is interesting and contains many information.

Major points

1. The authors performed a behavioral study using *Drosophila*. In my understanding, *Drosophila* does not have cilia in the brain, except for the type-I sensory neurons. Therefore, the role of cc2d1a in the role of antisocial behavior requires further explanation. Although these data are interesting, I do not think the data fit well in the manuscript, and therefore I suggest the authors to either move the data to a supplemental figure or add more data to show how cc2d1a affects the *Drosophila* brain. Does cc2d1a affect the morphology of brain and the function of neurons? In addition, I recommend the authors to discuss interpretation of the data in the discussion since cilia may not be involved in this phenotype.
2. In the methods section, the authors showed 90% and 81% knockout score for CRISPR/Cas9 injected embryos. Please describe which stage of the embryo was analyzed for genomic DNA extraction. Since it is known that F0 embryos can be mosaic, containing a mixture of wild-type and gene-edited alleles, I wonder how the authors managed to get such a high deletion score.

Minor comments.

1. Some characters are too small to read, such as the x and y axes in figure 2C, D and the scale bar (5um) in figure 2H, and more. Make them readable.
2. In figure 4F, gamma tubulin is not visible in a low mag image.
3. In figure 5A, what is the control? Identify this control in the methods section. Is this control a patient-related sample or a commercially purchased one?
4. Please provide the catalogue number for each item used in the manuscript.
5. The authors used "knockdown" instead of "knockout" with CRISPR/Cas9 *xenopus*. I thought knockdown is used for morpholino or siRNA, and knockout is used for CRISPR/Cas9.
6. Please include the information of PKD1 mutation in the table1
7. In figure 4a legend, please describe the time frame bar shown in the figure.
8. In the abbreviation section, there are more abbreviations used in the manuscript. List them all. I do not recommend using "ID" for intellectual disability.

Reviewer #2 (Comments to the Authors (Required)):

This is an interesting and well written paper describing 2 families with homozygous nonsense mutations on CC2D1A. The clinical phenotypes are reviewed and compared to known phenotypic spectrum. Animal models and patient fibroblasts are then used to show the defect in l-r axis, csf ciliary motility and primary cilia defects. I am supportive of the extension of clinical phenotypes associated with the mutations but additional data to show known and published phenotypes alongside the cases presented would be helpful. One of the probands, the cystic kidney phenotype is confounded by a de novo pkd1 allele but this is dealt with well. The animal data is robust and shows mechanistically how some of the disease phenotypes are devolved. The human fibroblast data is interesting, but the cilia figures are of low quality and should be improved.

REBUTTAL**Reviewer#1****MAJOR**

1. The authors performed a behavioral study using *Drosophila*. In my understanding, *Drosophila* does not have cilia in the brain, except for the type-I sensory neurons. Therefore, the role of *cc2d1a* in the role of antisocial behavior requires further explanation. Although these data are interesting, I do not think the data fit well in the manuscript, and therefore I suggest the authors to either move the data to a supplemental figure or add more data to show how *cc2d1a* affects the *Drosophila* brain. Does *cc2d1a* affect the morphology of brain and the function of neurons? In addition, I recommend the authors to discuss interpretation of the data in the discussion since cilia may not be involved in this phenotype.

We appreciate the reviewer's insightful comment about cilia being expressed only in type-I sensory neurons in *Drosophila* and the potential need for further studies to explain the antisocial behavior in flies.

We arrived at this approach by noting that mutations in RFX family transcription factors cause autism, attention-deficit disorder, intellectual disability, and dysregulated behavior (1). In *Drosophila*, Rfx mutants exhibit defects in chemosensory and mechanosensory behavior, and ultrastructure analysis shows that loss of Rfx is associated with a severe disorganization of the cilia at the tip of the sensory neuron dendrites (2), suggesting an association of ciliary defects with these neurodevelopmental phenotypes. Our studies in *Xenopus* showed the expression of *cc2d1a* in a variety of ciliated structures, including otic vesicles, ependyma, left-right organizer, kidneys, and cultured fibroblast cells from our patients showed defective Type1 cilia formation. Given that loss of Rfx in flies led to behavioral defects and cilia disorganization, we postulated that loss of *cc2d1a* might lead to a similar phenotypes. As one of the goals of our lab is to dissect the role of cilia in the brain and neurodevelopment, we performed the behavioral experiments in flies. However, we agree with the reviewer that the rigor warranted to study the role of *cc2d1a* on cilia in sensory neurons in flies is beyond the scope of the present report. Therefore, as recommended, we moved the *Drosophila* behavior assay results to supplementary data.

1. Disruption of RFX family transcription factors causes autism, attention-deficit/hyperactivity disorder, intellectual disability, and dysregulated behavior. Holly K. Harris et al. *Genetics in Medicine* (2021) 23:1028 – 1040 (<https://www.nature.com/articles/s41436-021-01114-z>)

2. *Drosophila* Regulatory factor X is necessary for ciliated sensory neuron differentiation
Raphaelle Dubruille, et al. *Development* (2002) 129 (23): 5487-5498
(<https://journals.biologists.com/dev/article/129/23/5487/41851/Drosophila-Regulatory-factor-X-is-necessary-for>).

2. In the methods section, the authors showed 90% and 81% knockout score for CRISPR/Cas9 injected embryos. Please describe which stage of the embryo was analyzed for genomic DNA extraction. Since it is known that F0 embryos can be mosaic, containing a mixture of wild-type and gene-edited alleles, I wonder how the authors managed to get such a high deletion score.

We apologize for the lack of clarity on our data and thank the reviewer for the attention to detail. The reviewer is correct; the F0 embryos are mosaic, and the 90 and 81% knockout score only reflects two individual experiments. We achieved knockout scores **up to** 90%(sgRNA#1) and 81% (sgRNA#2). We have modified the methods section (**CRISPRs, mRNA, and injections**) to clarify the text. We also analyzed additional tadpoles at stage 46 (4 days post-fertilization at 26°C) and expanded the Supplemental Figure 1 to show our cumulative data on CRISPR editing.

We use multiple steps to improve CRISPR efficiency. We start by producing sgRNAs using the predictive sgRNA-scoring algorithm CRISPRscan, built with the support of the *Xenopus* lab here at Yale (Giraldez and Khokha Lab) (3). CRISPRscan identifies potential CRISPR cut sites, designs oligos for sgRNA synthesis, and provides critical information on the frequency of cut sites in the genome (potential off-target effects). We select sgRNAs that cut at the 5' end of genes and are not predicted to cut elsewhere in the genome. We also use Cas9 protein (PNA Bio) rather than Cas9mRNA, which has better cutting efficiency and less toxicity in our hands. Before each injection, we incubate the sgRNA+Cas9 mixture for 10 min at 37°C to create nucleoprotein complexes to improve efficiency. Then, we inject the sgRNA+Cas9 protein mixed with either mRFP or Alexa 488 to visualize and confirm the injection accuracy. With these steps, we often reach >60% editing efficiency. If we do not, then we design additional sgRNAs.

3. Moreno-Mateos MA, Vejnar CE, Beaudoin JD, Fernandez JP, Mis EK, Khokha MK, Giraldez AJ. CRISPRscan: designing highly efficient sgRNAs for CRISPR-Cas9 targeting in vivo. Nat Methods. 2015;12(10):982-8. Epub 2015/09/01. doi: 10.1038/nmeth.3543. PubMed PMID: 26322839; PMCID: PMC4589495.

Minor

1. Some characters are too small to read, such as the x and y axes in figure 2C, D and the scale bar (5um) in figure 2H, and more. Make them readable. - 2. In figure 4F, gamma tubulin is not visible in a low mag image. - 6. Please include the information of PKD1 mutation in the table1 - 7. In figure 4a legend, please describe the time frame bar shown in the figure.

We improved our figure fonts and embedded images for better presentation.

3. In figure 5A, what is the control? Identify this control in the methods section. Is this control a patient-related sample or a commercially purchased one?

As the patient cells are primary fibroblasts, we utilized primary fibroblasts from a healthy, unrelated donor as a control. This has now been noted in the Methods section as recommended.

4. Please provide the catalogue number for each item used in the manuscript.

We apologize for the oversight; we have now included all the catalog numbers.

5. The authors used “knockdown” instead of “knockout” with CRISPR/Cas9 xenopus. I thought knockdown is used for morpholino or siRNA, and knockout is used for CRISPR/Cas9.

We corrected the text accordingly and replaced knockdown with either 'knockout' or 'depletion.'

8. In the abbreviation section, there are more abbreviations used in the manuscript. List them all. I do not recommend using "ID" for intellectual disability.

We expanded the abbreviations and replaced ID with IDD: Intellectual and Developmental Disabilities.

Reviewer #2

1. I am supportive of the extension of clinical phenotypes associated with the mutations but additional data to show known and published phenotypes alongside the cases presented would be helpful.

We thank the reviewer for the suggestions for improving our manuscript. We added additional table (Table 1), including the known and published phenotypes.

2. The human fibroblast data is interesting, but the cilia figures are of low quality and should be improved.

We now forwarded the higher resolution images to improve the quality

July 19, 2024

RE: Life Science Alliance Manuscript #LSA-2024-02708R

Dr. Engin Deniz
Yale School of Medicine
Pediatric Critical Care
333 Cedar st
New Haven 6520

Dear Dr. Deniz,

Thank you for submitting your revised manuscript entitled "CC2D1A causes ciliopathy, intellectual disability, heterotaxy, renal dysplasia and abnormal CSF flow". We would be happy to publish your paper in Life Science Alliance pending final revisions necessary to meet our formatting guidelines.

- please be sure that the authorship listing and order is correct
- please upload your Tables in editable .doc or Excel format
- please add ORCID ID for the secondary corresponding author -- they should have received instructions on how to do so
- please add the Twitter handle of your host institute/organization as well as your own or/and one of the authors in our system
- please create an email address for MarmaRare Group and add it to the system as a coauthor
- please consult our manuscript preparation guidelines <https://www.life-science-alliance.org/manuscript-prep> and make sure your manuscript sections are in the correct order
- we encourage you to revise the figure legends for Figure 3 such that the figure panels are introduced in alphabetical order
- please add callouts for Figures 1C; 3F, G; 5F; S1A-B; S2A-C; S3A-F and S4A-D to your main manuscript text
- please consider uploading the new variant information to ClinVar

FIGURE CHECKS

- please add sizes next to blots in Figure 5A
- please add scale bars to Figure 5D-F

A. FINAL FILES:

B. MANUSCRIPT ORGANIZATION AND FORMATTING:

Sincerely,

July 30, 2024

RE: Life Science Alliance Manuscript #LSA-2024-02708RR

Dr. Engin Deniz
Yale School of Medicine
Pediatric Critical Care
333 Cedar st
New Haven 6520

Dear Dr. Deniz,

Thank you for submitting your Research Article entitled "CC2D1A causes ciliopathy, intellectual disability, heterotaxy, renal dysplasia and abnormal CSF flow". It is a pleasure to let you know that your manuscript is now accepted for publication in Life Science Alliance. Congratulations on this interesting work.

DISTRIBUTION OF MATERIALS:

Again, congratulations on a very nice paper. I hope you found the review process to be constructive and are pleased with how the manuscript was handled editorially. We look forward to future exciting submissions from your lab.

Sincerely,
